# DMA, a Small Molecule, Increases Median Survival and Reduces Radiation-Induced Xerostomia via the Activation of the ERK1/2 Pathway in Oral Squamous Cell Carcinoma

**DOI:** 10.3390/cancers14194908

**Published:** 2022-10-07

**Authors:** Palak Parashar, Monoj Kumar Das, Pragya Tripathi, Tejinder Kataria, Deepak Gupta, Deepak Sarin, Puja Panwar Hazari, Vibha Tandon

**Affiliations:** 1Special Centre for Molecular Medicine, Jawaharlal Nehru University, New Delhi 110067, India; 2Division of Radiation Oncology, Medanta―The Medicity, Gurgaon 122001, India; 3Head and Neck OncoSurgery, Medanta―The Medicity, Gurgaon 122001, India; 4Defence Research and Development Organization, Institute of Nuclear Medicine and Allied Sciences, Delhi 110054, India

**Keywords:** head and neck, patient-derived xenograft, salivary gland cytoprotectant, xerostomia

## Abstract

**Simple Summary:**

Radiotherapy is commonly used to treat the majority of patients with head and neck cancers. Salivary glands in the radiation field are affected by this procedure. The purpose of this study was to investigate the role of DMA as a radiomodulator to evaluate the real possibilities of reducing the incidence and severity of xerostomia in head and neck squamous cell carcinoma (HNSCC) patients. The effect of DMA on the response of head and neck cell lines in the presence of radiation was analyzed using clonogenic survival and xenograft tumor assays for cell line-derived xenografts as well as patient-derived xenografts. The combinatorial treatment of DMA along with radiation influenced the migration of cells. The changes associated with migration were observed through a microscope and wound healing assay. In addition, the stimulated saliva was measured. Tissue sections were analyzed using immunohistochemistry to stain for molecular markers of cell proliferation and porin channels.

**Abstract:**

Survival, recurrence, and xerostomia are considerable problems in the treatment of oral squamous carcinoma patients. In this study, we investigated the role of DMA (5-(4-methylpiperazin-1-yl)-2-[2′-(3,4-dimethoxyphenyl)5″benzimidazoyl]benzimidazole) as a salivary gland cytoprotectant in a patient-derived xenograft mouse model. A significant increase in saliva secretion was observed in the DMA-treated xenograft compared to radiation alone. Repeated doses of DMA with a high dose of radiation showed a synergistic effect on mice survival and reduced tumor growth. The mean survival rate of tumor-bearing mice was significantly enhanced. The increased number of Ki-67-stained cells in the spleen, intestine, and lungs compared to the tumor suggests DMA ablates the tumor but protects other organs. The expression of aquaporin-5 was restored in tumor-bearing mice injected with DMA before irradiation. The reduced expression of αvβ3 integrin and CD44 in DMA alone and DMA with radiation-treated mice suggests a reduced migration of cells and stemness of cancer cells. DMA along with radiation treatment results in the activation of the Ras/Raf/MEK/ERK pathway in the tumor, leading to apoptosis through caspase upregulation. In conclusion, DMA has strong potential for use as an adjuvant in radiotherapy in OSCC patients.

## 1. Introduction

Nasopharyngeal carcinoma (NPC) is a rare head and neck cancer with a ratio of 0.6–2.0/100,000 in males and 0.2–0.8/100,000 in females worldwide [1]. However, there is significant variation in the geographic distribution of the disease; the highest incidence is found in Southeast Asia, with up to 6.4/100,000 males and 2.4/100,000 females in these regions. India, while an integral part of Southeast Asia, has significant geographic, racial, and cultural diversity in its population, which is reflected in the varied incidence of cancer in various parts of the country [2]. The highest age-adjusted rates for cancer were found in the northeast states, with the Kohima district in Nagaland having an incidence of 19.4/per 100,000 population [3]. While breast, colorectal, pancreas, prostate, and lung cancer are the major cancers in the developed world, head and neck cancer is the predominant sub-site in India and constitutes 25–30% of all cancers, as opposed to 3–4% in the Western world. The major cause of HNSCC in India is tobacco consumption, and clinicians are of the opinion that we need to study new diagnostic and prognostic markers among Indian patients and compare them with those from the Western world to develop tailored treatment regimens [4]. Radiotherapy can effectively mitigate head and neck cancer, either as a primary treatment or postsurgical adjuvant modality [5]. However, damage to the salivary glands caused by radiation often induces xerostomia [6], a dryness of the oral cavity resulting from insufficient saliva secretion, or a complete lack of saliva [7]. Saliva plays a vital role in maintaining oral health; water is its main component, constituting 99% of its volume [8]. Radioprotectors are compounds that selectively protect normal tissues over cancerous tissues and improve the therapeutic window of radiation therapy. The flagellin-derived polypeptide CBLB502, which binds to Toll-like receptor 5 (TLR5) and activates nuclear factor kappa light chain enhancer of activated B cells (NF-κB), enterocytes, and intestinal endothelial cells, is both a radioprotector and a mitigator of radiation injury in mouse models [9]. Oral pilocarpine and cevimeline are the only medications approved by the United States Food and Drug Administration (FDA) to treat xerostomia in patients with head and neck cancer [10]. Pilocarpine is a muscarinic cholinergic agonist that stimulates transient salivary secretion [11]. The first trial with pilocarpine for radiation-induced xerostomia was performed during the early 1990s. Pilocarpine significantly improved oral dryness in approximately half of the patients treated for 8–12 weeks at doses of >2.5 mg administered three times daily. However, its use is contraindicated for patients with asthma, acute iritis, or glaucoma; thus, pilocarpine should be used with extreme caution in patients with chronic obstructive pulmonary disease and cardiovascular disease [12]. Cevimeline is also effective in treating xerostomia in patients with Sjögren’s disease and shows potential for the treatment of radiation-induced xerostomia [13]. Other drugs, including bethanechol, methacholine, and carbachol, have given inconclusive results [14]. Hyperbaric oxygen has been shown to increase salivary function in patients who have undergone radiation [15]. In a phase III trial by Brizel et al. [16], the daily intravenous administration of amifostine before each fraction (200 mg/m^2^) significantly reduced the incidence of acute xerostomia without altering the survival rate. The most common side effects of amifostine are transient hypotension, nausea, and emesis [17]. Palifermin, a truncated human recombinant keratinocyte growth factor generated in *Escherichia coli*, upregulates cytoprotective mechanisms. It has been shown to be effective in minimizing the severity of mucositis in head and neck cancer patients being treated with chemotherapy and radiotherapy [18]. However, none of these strategies have been uniformly successful, and the treatment of mucositis remains an unmet need. There is an urgent need for a potent radioprotector for HNSCC patients. DMA is a synthetic molecule synthesized in our laboratory [19] that affords significant protection against radiation-induced damage to normal cells and exhibits effective whole-body radioprotection at 1/7th the dose of its maximum tolerable dose (MTD) of 2000 mg/kg body weight (bw) orally [20]. It has been previously shown that DMA induces NIK-mediated NFκB activation and modulates several key regulatory pathways to overcome radiation-induced damage in vitro [21]. Here, we deciphered the role of DMA as a salivary gland cytoprotectant in tumor-bearing Crl:NU(NCr)-*Foxn1^nu^* Athymic nude mice against focal radiation in a cell line-derived xenograft and a patient-derived xenograft (PDX) model.

## 2. Materials and Methods

The source, catalog number, and unique research resource identifier (RRID) code of the antibodies, software, chemical, and reagents used in this study are listed in Appendix A.

### 2.1. Ethics Approval and Study Subjects

The Institutional Review Board of Medanta, The Medicity, Gurugram, Haryana, India approved the study (No. MICR-1013/2019). The study was conducted following the ethical guidelines of the declaration. Oral squamous cell carcinoma (OSCC) tumor samples were collected from Indian patients receiving treatment at Medanta hospital in Gurugram [22]. The histology of the specimens was interpreted by certified pathologists, and the tumor staging was determined according to the American Joint Committee on Cancer (AJCC) staging manual, 7th edition.

### 2.2. Cell Lines

Nasopharyngeal HK-1 (CVCL_7084) and hypopharyngeal FaDu (HTB-43) cells were purchased from the American Type Culture Collection (ATCC). HNMT1 primary cells were derived from a 65-year-old male patient with tongue squamous cell carcinoma. The patient tissue sample was placed in a culture medium immediately after surgical resection. The tissue samples were transported directly to the laboratory for the preparation of single cells at room temperature. The tissue was minced in minimum Eagle’s medium (EMEM) into a fine slurry with a scalpel to obtain 3mm^3^ pieces. Collagenase (200 U/mL) and DNase (50 U/mL) in Hank’s balanced salt solution (HBSS) was added to the tissue and mixed with a serological pipette. The tube was kept at 37 °C for 15 min and tapped 20 times every 5 min. The digested tissue sample was centrifuged at 1000 rpm for 4 min. The supernatant was discarded by pipetting and HBSS was added, followed by centrifugation at 1000 rpm for 4 min. The supernatant was discarded and TEG (0.025% trypsin, 40 µg/mL ethylene glycol-bis(β-aminoethyl ether)-N,N,N′,N′-tetraacetic acid (EGTA), and 50 µg/25 µL polyvinyl alcohol) was added to the tube. The tube was kept at 37 °C for 30 min. The cells were strained with staining media (1% bovine serum albumin, HEPES, and 1 × penicillin–streptomycin antibiotic) using a 40 µm nylon cell strainer. The collected strained cell suspension was centrifuged at 900 rpm for 5 min, resuspended in EMEM, and counted to quantify viable cells using trypan blue [23]. Short tandem repeat (STR) DNA profiling of the patient-derived primary cell line was performed from the DNA extracted using a DNA Miniprep Kit. The profiles were compared to the STR profiles database [24].

Roswell Park Memorial Institute medium (RPMI-1640) was used for the HK-1 cells, while minimum essential Eagle’s Medium (EMEM) was used for the FaDu and HNMT1 cells with 10% fetal bovine serum (FBS) and 1X antimycotic antibiotic solution. No Mycoplasma contamination was detected when using the Look Out Mycoplasma PCR detection kit every 6 months. All cell lines used for experiments were not cultured for more than 30 passages.

### 2.3. Patient-Derived Primary Cells—Acquisition and Sequencing

The sequenced raw data from HNMT1 primary cells were processed to obtain high-quality reads using fastp to remove the adapter sequences and reads with more than a 10% quality threshold (QV) and Phred score <20. After removing the adapter and low-quality sequences from the raw data, high-quality reads were obtained (Appendix A, filename: fastp report). The RNA-seq paired-end sequencing library was prepared from the high-quality reads using Illumina TruSeq (San Diego, CA, USA). After obtaining the Qubit concentration for the library, the paired-end Illumina library was loaded onto NextSeq500 for sequencing [25]. These high-quality paired-end reads were mapped to the reference genome of Homo sapiens (hg19) using the spliced transcripts alignment to a reference (STAR) expression count (Appendix A, filename: HNMT1.hg19.bam.bai). STAR has been shown to have high accuracy as it is substantiated by an algorithm that functions using seed searching followed by clustering, stitching, and scoring.

### 2.4. In Vitro Cytotoxicity Assay

A cytotoxicity evaluation of DMA was performed using a 3-(4,5-Dimethylthiazol-2-yl)-2,5-diphenyltetrazolium bromide (MTT) assay [26]. Briefly, HK-1, FaDu, and HNMT1 cells were harvested, counted, and seeded at 3 × 10^3^ cells per well in 96-well plates for 24 h. The following day, the cells were treated with various concentrations of DMA and incubated for 24, 48, and 72 h at 37 °C with 5% CO_2_. After that, 0.5 mg/mL of MTT was added to each well and incubated for 4 h. Next, the solution was removed and 150 µL of dimethyl sulfoxide (DMSO) was added to the wells. Then, the plate was read at 575 nm by an infinite M200 pro. The results were analyzed as the percentage of viable cells to the concentration of DMA.

### 2.5. DMA, AMG 510, and Radiation Exposure

DMA was synthesized in our laboratory, as shown in Figure 1A [19], and its chemical structure is depicted in Figure 1A. HK-1, FaDu, and HNMT1 (derived from Medanta head and neck patient 1) cells were exposed to 2, 4, 6, and 8 Gy of X-ray irradiation (1.2 Gy/min) using a Precision X-RAD 225 (Advanced Instrumentation Research Facility, Jawaharlal Nehru University, Delhi, India). Focal irradiation at 10, 20, and 40 Gy was given to mice at the tumor site using a field size of 20 × 20 cm at a 50 cm source-to-surface distance (SSD), 225 kilovolts (kV), and 13.3 milliamperes (mA). AMG 510 (Sotorasib) is a selective KRAS G12C covalent inhibitor. It irreversibly inhibits KRAS G12C by locking it in an inactive GDP-bound state. It was reported that treatment with AMG 510 resulted in the suppression of ERK phosphorylation in different cancer cells; however, as with KRAS silencing, it resulted in a significant increase in the phosphorylation of AKT on Ser-473. Hence, AMG 510 is now being used as an ERK1/2 inhibitor also [27,28].

### 2.6. Clonogenic Survival Assay

HK-1, FaDu, and HNMT1 cells at a density of 0.5 × 10^6^ were treated with DMA, irradiated at 2, 4, 6, and 8 Gy, seeded into 6-well plates at different densities, and allowed to grow for 7–14 days until colony formation at ≥50 cells. The colonies were fixed, stained with crystal violet, and counted. Survival fractions were corrected for each group with the untreated control survival fraction. Radioprotection was expressed as a dose modifying factor (DMF) calculated as the ratio of mean inactivation dose (DMA + radiation)/mean inactivation dose (radiation) [29].

### 2.7. Wound Healing Assay

The migratory activity of HK-1 cells was studied using a wound healing assay, as previously reported [30]. HK-1 cells (3 × 10^5^) were grown in triplicates under normal growth conditions in six-well plates for the following four experimental groups: control, DMA (10 and 15 μM), radiation (4 Gy), and DMA + radiation (10 μM + 4 Gy and 15 μM + 4 Gy). The cells were treated with DMA for 2 h followed by radiation, then they were washed and incubated with the corresponding cell medium. Then, a narrow area on the confluent cell monolayers was scratched off with a p200 pipette tip. The cells were allowed to migrate for 24, 48, and 72 h, and images from the same areas were obtained with an inverted microscope. The images were analyzed with the Image J software (National Institutes of Health, Bethesda, MD, USA). For each image, the cell migration rate was compared from 0 to 24, 48, and 72 h using the following formula: cell migration distance/migration time [31].

### 2.8. Atomic Force Microscopy (AFM)

The ultrastructure of HK-1 cells grown in six-well plates for four experimental groups—control, DMA (10 and 15 μM), radiation (4 Gy), and DMA + radiation (10 μM + 4 Gy and 15 μM + 4 Gy)—was examined using AFM. The cells were treated with DMA for 2 h followed by radiation. Before imaging, cells cultured on glass coverslips were washed three times with PBS to remove dead or non-adherent cells, and serum-free medium was added to six-well plates. A sharpened silicon nitride cantilever tip was used for cell scanning. The cantilever sensitivity was calibrated by indenting the glass substrate in the presence of a cell culture medium. A representative optical microscope screenshot was obtained using AFM. All indentations were performed on randomly selected single cells [32].

### 2.9. Animal Procedures

Briefly, 4 to 5-week-old CrI: NU (NCr)-Foxn^nu^ female/male mice were housed at 50 ± 5% relative humidity and 25 ± 2 °C under pathogen-free conditions at Jawaharlal Nehru University (Registration No. 19/GO/ReBi/99/CPCSEA dated 10 March 1999) as per the recommendation of the Committee for the Purpose of Control and Supervision of Experiments on Animals (CPCSEA), India and Animal Research: Reporting of In Vivo Experiments (ARRIVE) guidelines in a single-blind study.

### 2.10. Generation of Xenograft Model and Therapy Response Experiments

An HK-1 xenograft was developed in mice. Briefly, an over 90% viable cell suspension was prepared in incomplete RPMI-1640 medium and a 1:1 ratio of Matrigel^®^. Tumors were generated by injecting 100 µL of cell (3 × 10^6^) suspension subcutaneously into one side of the back of the mice. The development of the patient-derived xenograft (PDX) model was carried out using the tumor tissues from three HNSCC patient samples—HNMT1 (aged 65 years, male, site of cancer in the tongue), HNMT2 (aged 36 years, male, cancer in the upper alveolus), and HNMT3 (aged 49 years, male, cancer in the lower alveolus)—collected from Medanta Hospital, Gurgaon, in medium 199 (M199) media supplemented with 10% FBS and 1% antimycotic antibiotic solution. The effect of DMA was evaluated in the PDX of three different patient samples; however, the results from only one group are shown here. The trend of the results was similar between the three PDX mouse models. The tumor tissue was cut into 3 × 3 × 3 mm pieces and implanted into nude mice (F_0_ generation). Using a scalpel blade, an approximately 5 mm long incision was made on the back of the mice to create a subcutaneous pocket. The wound edges were closed by applying Vetbond 3M tissue adhesive. After one week, tumor volume and body weight were monitored, and upon reaching ~1500–2000 mm^3^, the tumors were transplanted into new mice (F_1_ generation). Similarly, the F_2_ generation was grown as shown in Appendix A [33]. Just before treatment initiation, the animals were randomized into treatment groups of six mice each. The treatment groups for the HK-1 xenograft were control (untreated), amifostine (200mg/kg), DMA (50 mg/kg), radiation (4 Gy), amifostine + radiation (200 mg/kg + 4 Gy), and DMA + radiation (50 mg/kg + 4 Gy) and for PDX were control (untreated), DMA (25 mg/kg), radiation (4 Gy), and DMA + radiation (25 mg/kg + 4 Gy). DMA and amifostine were intravenously injected for five consecutive days, followed by 4 Gy of fractionated focal X-ray irradiation at the tumor site. In addition to body weight, tumor volume was measured every other day using a digital Vernier caliper and determined using the following formula: 0.5 [length] [width]^2^. The successful generation of the PDX was further checked by aligning reads with quality scores for the F_0_ and F_1_ generations using STAR (Appendix A).

### 2.11. Determination of Xerostomia by Measuring Saliva Produced in All Experimental Groups after Radiation Treatment

Mice were orally injected with 2.5 mg/mL pilocarpine HCl (P6503-5G); after 10 min, saliva was collected using a 200 µL micropipette at 25 °C and stored in 1.5 mL microcentrifuge tubes. Saliva secretion was calculated as the volume of saliva (µL) produced upon stimulation with pilocarpine. Saliva was collected on the first, third, and fourth week for the HK-1 nasopharyngeal xenograft, while for the PDX, it was collected on the first, third, fifth, and seventh week, and normalized to the control mice group to determine the differential effect of DMA on the production of saliva with radiation treatment [34]. The saliva volume was pooled for each group with six mice, and statistical significance was estimated using GraphPad Prism 8. The results were assayed using two-way ANOVA followed by Tukey’s multiple comparisons posthoc analysis. Water consumption was also recorded each day after drug and radiation treatment and throughout the study.

### 2.12. Mathematical Determination of Synergy

The synergy between DMA and radiation treatment was determined based on the Bliss definition of drug independence. Briefly, the individual curves of tumor growth delay (TGD) for each animal in a treatment cohort were log-transformed and fitted using linear regression. The slope values were then placed into the following equation: [Radiation] + [DMA] − [Radiation and DMA combination] − [Control]. If the final value was greater than 1.0, then the combined response was considered synergistic [35].

### 2.13. Histological Studies of Tissue Sections of HK-1 Xenograft and PDX MouseModels

The tumor and the organs, including the intestines, lungs, and spleen, were collected after sacrifice, washed with PBS, soaked in blotting paper, and immediately fixed in 10% neutral formalin buffer. The formalin-fixed tissues were trimmed and dehydrated using increasing grades of alcohol (70%, 95%, and 100%), cleared with xylene, and finally infiltrated with paraffin. The tissues were sectioned into 5μm pieces using a rotary microtome (Leica RM2245, Wetzlar, Germany) and stained with hematoxylin and eosin (H&E). The detailed methodology is discussed in Appendix A. The images were captured using a Nikon TE-2000S (Tokyo, Japan) microscope and a DSFi3 camera and compared with the images of the normal tissues from the same mice.

### 2.14. Immunohistochemistry of AQP5, Ki-67, and CD44-Positive Cells in HK-1 Xenograft and PDX MouseModels

Tissue sections from the tumor, lungs, small intestine, and spleen samples (three animals/group) were evaluated for Ki-67 expression, the salivary gland was assessed for the expression of aquaporin 5 (AQP5), and the tumor tissue for CD44 by randomly selecting three fields per tissue section. The detailed methodology is discussed in Appendix A.

### 2.15. Immunohistochemistry of Integrin β3 in PDX MouseModel

Immunohistochemistry for integrin β3 was performed using a PolyExcel DAB detection system by PathnSitu, Biotechnologies Pvt. Ltd., Hyderabad, Telangana, India. The detailed methodology is discussed in Appendix A.

### 2.16. ^99m^Tc Single-Photon Emission Computed Tomography (SPECT) Imaging

Mice were anesthetized using 60 µL of ketamine and xylazine cocktail followed by the intravenous injection of 100µCi Technetium-99m (^99m^Tc) radionuclide agent-labeled arginine–glycine–aspartate (RGD) peptide. Two hours post injection, a whole-body image was acquired using a Symbia T6 SPECT. The SPECT emission image data were processed using ordered subset expectation maximization reconstruction software with two iterations and eight subsets [36].

### 2.17. Western Blot Analysis of Protein Lysate fromPDX Tumor Tissue

The tumor tissues of the patient-derived xenografts treated with 4 Gy irradiation, 25 mg/kg DMA, and 25 mg/kg DMA along with 4 Gy for 5 consecutive days were lysed using RIPA buffer. Proteins were quantified using Bradford reagent. A total of 60 µg of protein was loaded, separated using SDS-PAGE, and transferred on to a PVDF membrane (Millipore) using wet transfer. Western blotting, including blocking and probing with antibodies, was performed according to the manufacturer’s instructions. The protein for Ki-67 was separated by SDS-PAGE at 100 V for 5 h and wet transferred onto a PVDF membrane at 30 V overnight. The antibody was used as per the manufacturer’s instructions, with three primary and secondary washing cycles for 5 min each with 1 × TBST. A detailed list of antibodies is provided in Appendix A. The blots were developed using Clarity^TM^ ECL chemiluminescent solution either on X-ray film or in a ChemiDoc (Biorad, Hercules, CA, USA).

### 2.18. Annexin V-FITC Apoptosis Assay for the Detection of the Apoptotic Cells

HK-1 cells grown in six-well plates for the control, DMA (6.25 µM), radiation (4 Gy), DMA + radiation (6.25 µM + 4 Gy), AMG510 (10 µM), AMG510 + radiation (10 µM + 4 Gy), and DMA + AMG510+ radiation (6.25 μM + 10 μM + 4 Gy) groups. The cells were washed twice with cold PBS and then resuspended in 1× binding buffer at a concentration of 1 × 10^6^ cell/mL. A volume of 100 µL of the solution (1 × 10^5^ cell) was transferred to a 5 mL culture tube. A total of 5 µL of annexin V-FITC and 5 µL of propidium iodide was added to the cells, followed by incubation for 15 min at 25 °C in the dark. Binding buffer (1×) was added to each tube and analyzed using a BD LSR Fortessa within one hour.

### 2.19. Hematology Analysis of HK-1 Xenograft MouseModel

Three mice out of each group—control, DMA (50 mg/kg), radiation (4 Gy), and DMA + radiation (50 mg/kg + 4 Gy)—were used for blood collection (100 µL) from the ventricle, accessed via the left side of the chest. Blood samples were taken without an anticoagulant in a sterile 1.5 mL microcentrifuge tube for the separation of serum, which was used to measure biochemical parameters. The detailed methodology is discussed in Appendix A.

### 2.20. Statistical Analysis

The Kaplan–Meier estimator was used to generate survival curves and to estimate the median survival in the HK-1 xenograft and PDX mouse models. All experiments were performed at least thrice. Statistical significance was estimated using GraphPad Prism 8. Data were analyzed using either one-way or two-way ANOVA followed by Tukey’s multiple comparison post hoc analysis. The results of the experiments with only two parameters were compared using an unpaired Students t-test. *p*-values at * < 0.05, ** < 0.005, *** < 0.0005, and **** < 0.00005 indicate statistical significance.

## 3. Results

### 3.1. DMA Does Not Protect Head and Neck Cancer Cells and Primary Cells Derived from a Patient with Tongue Squamous Carcinoma against Radiation

The cytotoxicity of DMA in HK-1, FaDu, and HNMT1 cells was evaluated using an MTT assay. DMA exhibited significant cell-killing activity against head and neck cancer cell lines, with half-maximal inhibitory concentration (IC_50_) values of 6.25,11.33, and 36.138 µM, respectively, after 48 h of incubation with DMA (Figure 1B–D). In addition, we conducted an in vitro clonogenic survival assay using three head and neck cancer cell lines. The clonogenicity assay suggested a dose modification factor (DMF) of 0.932, 0.85, 0.79, and 0.725 for HK-1 (Figure 1E); 0.92, 0.85, 0.793, and 0.73 for FaDu (Figure 1F); and 0.96, 0.94, 0.91, and 0.886 for HNMT1 cells at 2, 4, 6, and 8 Gy, respectively (Figure 1G). No protection was observed across the DMA-treated head and neck cell lines, as according to the literature, the DMF value of a radioprotector should be in the range of 1.1 to 2.7 [37].

### 3.2. DMA Treatment Decreases Radiation-Induced Migration

The effect of DMA on cell migration was also observed. The motility of HK-1 was evaluated using a cell scratch assay. The results demonstrated that the mean scratch distance significantly decreased to 3.62, 3.83, and 4.1 µm in the DMA + radiation group, compared to 7.64, 7.73, and 8.69 µm in the group with radiation alone, after 24, 48, and 72 h of treatment, respectively (Figure 1H–K).

### 3.3. Morphological Changes in Nasopharyngeal Carcinoma Cells after Treatment with DMA

High-resolution AFM images of the control and treated HK-1 cells are depicted in Figure 1L. As can be seen from the figure, the control cells were characterized by a smooth surface and a well-defined nucleus. On the other hand, lamellipodia could be observed on the surface of the cells treated with radiation alone. The cell brushes were rarely seen in the HK-1 cells treated with DMA alone, while the group receiving the combinatorial treatment of DMA and radiation showed many small agglomerates around the treated cells, which probably originated from the fragmentation of the cell edges [38].

### 3.4. Salivary Gland Secretion Is Enhanced by the Systemic Delivery of DMA before Radiation in HK-1 Xenograft and PDX MouseModels

Head and neck cancer patients suffer from dryness of mouth caused by salivary gland dysfunction after radiotherapy [39]. Therefore, we measured the effect of DMA pretreatment on salivary gland function in the HK-1 xenograft and PDX mouse models after irradiation. The DMA and radiation treatments were given for five consecutive days to the mice, and saliva was collected weekly for 3–5 weeks post irradiation. The average saliva secretion in the HK-1 xenograft mice was 31.6, 14.6, 38.3, 32.1, 20.23, and 26.02 µL in the control, irradiated (4 Gy), DMA-treated (50 mg/kg), DMA (50 mg/kg) + 4 Gy-treated, amifostine 200 mg/kg, and amifostine 200 mg/kg + 4 Gy-treated groups, respectively. A significant increase in saliva secretion was observed in mice carrying the HK-1 xenograft in the DMA + radiation group as compared to the group treated with radiation alone (Figure 2A). We also observed that saliva secretion was improved in non-tumor-bearing mice to 15.14 and 14.81 µL for the DMA 50 mg/kg and DMA 50 mg/kg + 4 Gy groups, respectively, compared to 8 µL in the group receiving radiation alone (Figure 2B). A similar pattern was observed in terms of the restoration of saliva in the PDX model, with 15 and 12.73 µL of saliva in the DMA 25 mg/kg and DMA 25 mg/kg + 4 Gy groups, as compared to 3.6 µL in the group with radiation alone (Figure 3A). Water consumption was also normalized in the HK-1 xenograft and PDX mouse models, as shown in Appendix A.

### 3.5. DMA and Radiation Treatment Showed Synergistic Response Causing Significant Tumor Growth Delay in Nasopharyngeal and Patient-Derived Xenograft MouseModels

Radiation doses of 2,4, and 8 Gy were chosen for the HK-1 xenograft and PDX mouse models in the tumor growth delay (TGD) experiments to test for any potential for tumor radioprotection or radiosensitization. The notable antitumor effects of the combination seen in the HK-1 xenograft (Figure 2C) in contrast to the single-agent activity of DMA suggested additional mechanisms of radiation-induced tumor cell killing synergy in vivo that relied on high dose per fraction exposures. Similarly, in the PDX model, the effect of DMA + radiation on tumor growth was quite evident. With an increase in radiation dose, the tumor volume decreased to 136 mm^3^ in the DMA 25 mg/kg × 5 + 8 Gy × 5 groups compared to 180 mm^3^ in the 8 Gy × 5 experimental groups. The tumor volume in the 4 Gy × 5 and 2 Gy × 5 groups was 380 and 636 mm^3^, respectively, (Figure 3B). Based on the regression analysis, a synergistic response to the combination of DMA + 2 Gy (1.4), DMA + 4 Gy (2.02), and DMA + 8 Gy (1.3) was observed for the HK-1 xenograft. The response of the tumors in the PDX to radiation was enhanced in the DMA + 2 Gy (1.48), DMA + 4 Gy (1.37), and DMA + 8 Gy (1.088) groups. DMA showed a better response in reducing the tumor burden at high fractionated radiation doses (Figure 3C).

### 3.6. DMA Ameliorates Radiation-Induced Damage in Normal Tissues Compared to Tumor Tissue

The Crl: NU(NCr)-Foxn1 nude mice were sacrificed at the end of the study, and H&E staining of tumor tissue, spleen, and intestine was performed according to the standard protocol described in the “Histological studies of HK-1 nasopharyngeal HNSCC xenograft model” section. Extramedullary hematopoiesis in the spleen and the degeneration of the bronchiolar epithelium in the lungs were observed in the 4 Gy treatment group. However, the intestine of this group of animals did not show any adverse histopathological alteration. A clear decrease in the percentage of incidence of extra-medullary hematopoiesis in the spleen was observed in the DMA-treated group when compared to the radiation-exposed group. In the lungs, the percentage of incidence was not reduced; however, severity was found to be decreased when compared to animals exposed to radiation alone, indicating the protective effect of DMA. In the case of a tumor, the percentage of incidence of degeneration or necrosis in neoplastic cells remained constant between all the groups (Figure 4 and Appendix A).

Similarly, tumor tissues and other organs were also removed from the mice, and sections of the tissues from the four different groups of PDX mice were prepared and stained. The histopathological changes in the spleen, small intestine, and tumor of the PDX mice were observed. Extramedullary hematopoiesis in the spleen and amyloidosis in the small intestine were observed in the radiation-only treatment group. However, in the DMA + 4 Gy radiation treatment group, the animals did not show any adverse histopathological alteration in the spleen, and the level of incidence of amyloidosis along with autolysis in the small intestine was also around 11–25%. A decrease in the percentage of incidence of extra-medullary hematopoiesis in the spleen and amyloidosis in the small intestine was observed in DMA-treated group (around 26–50%) when compared to the radiation-exposed group (51–75%). In the case of a tumor, the percentage of incidence of degeneration or necrosis in neoplastic cells was high in the radiation-only group (Appendix A).

### 3.7. Restoration of Expression of AQP5 by DMA on the Apical Surface of the Salivary Gland in HK-1 Xenograft and PDX Models

AQP5 acts as the major apical water pathway for regulating water permeability in acinar cells and determines the flow rate and ionic components of secreted saliva [40]. The immunostaining of AQP5 was detected on the apical surface of acinar cells in the mouse models. The relative percentage of AQP5 density was higher for the 50 mg/kg DMA (96%) and 50 mg/kg DMA + 4 Gy radiation (94%) groups than for the 200 mg/kg amifostine (82%), 200 mg/kg amifostine + 4 Gy radiation (92%), and 4 Gy radiation (77%) groups in the HK-1 xenograft model. A similar pattern was observed in the PDX model for the relative percentage of AQP5 density (Figure 5A–C). Therefore, DMA with radiation treatment restored the expression of AQP5.

### 3.8. In Vivo Radioprotection by DMA in HK-1 Xenograft and PDX MouseModels

Ki-67is a large nuclear protein expressed during all active phases of the cell cycle and a marker of cellular proliferation [41]. The mean Ki-67 expression levels were 115 ± 12.13, 118 ± 47.18, 96 ± 15.56, and 151 ± 18.99 for the tumor, small intestine, spleen, and lungs, respectively, in the non-radiation control group in the HK-1 xenograft group of mice. Ki-67-positive proliferating cell numbers were found to be significantly reduced to 43 ± 18.31, 78 ± 14.35, and 73 ± 15.01 in the tumor, spleen, and lungs, respectively, in mice exposed to 4 Gy radiation compared to the non-radiation control groups. Pretreatment with the reference drug amifostine at a dose of 200 mg/kg before exposure to radiation at 4 Gy resulted in an increase in the number of proliferative cells to 94 ± 35.34 and 135 ± 11.67 in the intestine and lungs, respectively, as seen by an elevation in Ki-67-stained cells, but not in the tumor tissues, which had a mean expression level of 19 ± 8.83. A decreased number of Ki-67-stained cells was observed in the tumor tissue sections, similar to the radiation control groups. The tissue sections from the animals pretreated with DMA at 50 mg/kg followed by radiation exposure at 4 Gy per fraction showed an elevation in the number of Ki-67-stained proliferative cells to 212 ± 24.21, 109 ± 6.18, and 132 ± 5.29 in the small intestine, spleen, and lungs, respectively. The tumor tissue sections showed a decreased number of Ki-67-positive cells (41 ± 11.00) compared to the spleen, lungs, and intestine. These results suggest that DMA exerts radioprotection on normal tissues if injected just before irradiation (Figure 5D and Appendix A).

In the PDX model, the average number of Ki-67 positive cells in tumor tissue was reduced to 211 ± 8.2 in the DMA + radiation (25 mg/kg × 5 + 4 Gy × 5) group and 190.4 ± 9.3 in the radiation-only (4 Gy × 5) mice to as compared to the control (235.5 ± 3.5) group (Figure 5E). Hence, the present results suggest that DMA does not protect tumor tissue.

### 3.9. Quantification of Integrin α_v_β_3_ Expression Level in the Patient-Derived Xenograft Model

Integrin αvβ3 plays a role in tumor angiogenesis [42] and serves as a αβ heterodimeric receptor for extracellular matrix proteins with an exposed RGD tripeptide sequence. The expression of integrin αvβ3 during tumor growth, invasion, and metastasis presents an interesting molecular target for the diagnosis and treatment of rapidly growing and metastatic tumors [43]. The mean immunoreactivity scores for β3 integrins were significantly decreased in the DMA + radiation (201 ± 3.076) and DMA (211 ± 2.50) groups compared to the control (227.08 ± 2.81) and radiation (243.94 ± 2.56) groups, as depicted in the immunohistopathology of the tumor tissue (Figure 6A,B). In addition to the above, SPECT imaging of the tumor tissue from the patient-derived xenograft model of tongue squamous carcinoma was performed using a 100 µCi ^99m^Tc-labeled RGD peptide for the quantification of integrin αvβ3 levels (Figure 6C). The tumor uptake of labeled RGD peptide was expressed in terms of kilobecquerels (KBq), which reflected the total integrin αvβ3 level. For normalization, the values for tumor uptake (KBq) were divided by the tumor volume (mm^3^) of the four different treatment conditions—control, radiation, DMA, and DMA + radiation—to obtain the relative αvβ3 quantification values of 1.75, 0.58, 1.22, and 1.27 KBq/mm^3^, respectively (Figure 6D). As tumors grow, the total integrin αvβ3 level is higher, and tumor uptake increases. However, the integrin αvβ3 levels were lower in the radiation-only group, followed by the DMA, DMA + radiation, and control groups, as microvessel density decreased due to the maturity of blood vessels, which resulted in a decrease in integrin αvβ3 density because of a larger interstitial space. In addition, parts of the tumor may have become necrotic, leading to a lower integrin αvβ3 density [42].

### 3.10. Downregulation of the Expression of the Cancer Stem Cell Marker CD44 by DMA

CD44 is a transmembrane surface glycoprotein that normally functions as an adhesion molecule through its interactions with hyaluronan and cytoskeletal components in addition to maintaining tyrosine kinase activity [44]. The aberrant expression of CD44 in malignancy can lead to tumor extension and metastasis. We observed high CD44 expression in mice treated with 4 Gy radiation (241.18 ± 5.56), whereas weak CD44 expression was observed in the DMA (112.02 ± 5.37) and DMA + radiation (130.2 ± 6.32) groups, which was highly correlated with the morphological changes observed using AFM (Figure 6E,F).

### 3.11. DMA Induced Ras/Raf/MEK/ERK Pathway Leading to Caspase Activation to Initiate Apoptotic Signal in Tumor Cells

We observed the upregulation of Ras and ERK, which are involved in the Ras/Raf/MEK/ERK pathway along with cyclin B1, in DMA + radiation-treated tumors compared to those treated with irradiation only (Figure 7A). An increased expression of cleaved caspase-3 apoptotic proteins was observed in the DMA + radiation-treated tumors (Figure 7A). The Ki-67-positive cells were significantly reduced in the DMA + radiation group. The expression of the channel protein aquaporin 5 (AQP5) was increased significantly compared to the group treated with radiation only (Figure 7A,B).

### 3.12. Effect of DMA on the Apoptosis of Nasopharyngeal HK-1 Cells

Annexin V-FITC is used to determine the percentage of cells within a population that are actively undergoing apoptosis. Phospholipid phosphatidylserine (PS) in the membrane is translocated from the inner leaflet of the plasma membrane to the outer leaflet, thereby exposing it to the external environment. Annexin V has a high affinity for PS and is useful for identifying apoptotic cells with exposed PS. Propidium iodide (PI) is used to distinguish viable from nonviable cells, as viable cells with intact membranes exclude PI, whereas the membranes of dead and damaged cells are permeable to PI. Cells that stain positive for annexin V-FITC and negative for PI are undergoing apoptosis. Cells that stain positive for both annexin V-FITC and PI are either in the end stage of apoptosis, are undergoing necrosis, or are already dead. Cells that stain negative for both annexin V-FITC and PI are alive and not undergoing measurable apoptosis. The total percentage of the FITC+/PI- early apoptotic and FITC+/PI+ late apoptotic cell population was 3.31%, 81.87%, 95.1%, 46.34%, 45.8%, and 46.26% for the radiation (4 Gy), DMA (6.25 µM), DMA + radiation (6.25 µM + 4 Gy), AMG510 (10 µM), AMG510 + radiation (10 µM + 4 Gy), and DMA + AMG510 + radiation (6.25 µM + 10 µM + 4 Gy) groups, respectively. Based on our above findings, DMA induced apoptosis in the HK-1 cells (Figure 8).

### 3.13. Effect of DMA on the Biochemical Profile of the Xenograft

The biochemical profile of the blood of DMA-treated mice showed a reduction in the levels of AST and ALP enzymes (Appendix A). As per the literature, the AST levels of cancer patients have been found to be elevated, resulting in severe necrosis and neoplasia [45]. Our results suggest that AST levels were substantially reduced in the radiation, DMA, and DMA + radiation-treated mice as compared to the untreated controls, which indicates that DMA prevents hepatotoxicity and reduces cellular damage. ALP is an indicator of tumor proliferation, and its presence results in the rapid growth and relapse of disease [46]. Therefore, the DMA + radiation treatment mediated a reduction in the level of ALP compared to the control group of mice [47].

## 4. Discussion

Despite the discovery of several novel treatment modalities, radiotherapy remains the primary treatment for head and neck cancer. Our earlier in vitro studies illustrated DMA as a safe and effective radioprotective agent in normal lung fibroblast cells and human embryonic kidney cells [47]. Previously, it was also shown that a single dose of DMA administered orally in non-tumor-bearing mice protected against total body irradiation with a dose reduction factor (DRF) of 1.28 [48]. The DRF is the ratio of radiation doses required to produce the same biological effect in the absence and presence of the radioprotector [49]. DMA has a half-life of 3.5 h via oral administration and 2.65 h via *i.v.* administration and exhibits significant radioprotection [48] compared to amifostine, an FDA-approved radioprotector, which has a half-life of less than 1 min.

In addition to our previous findings showing that DMA treatment before the irradiation of melanoma-bearing mice did not provide radioprotection for the tumor [48], we highlight in this study that DMA does not protect head and neck cancer cells derived from tongue squamous cell carcinoma. On the contrary, we found a significant decrease in tumor volume in DMA-treated animals before irradiation. This implies that DMA caused the radiosensitization of tumors in vivo in the HK-1 xenograft and PDX mouse models.

Metastasis is the spread of cancer cells from the primary tumor to distant sites of the body to establish secondary tumors [50]. Our atomic force microscopy results showed that DMA affected the dynamics of the migration of nasopharyngeal HK-1 cells. The formation of filipodia-like structures, which guide cells to migrate from the site of infection to other target sites, was observed on the cellular membrane of HK-1 cells in the presence of radiation alone. In mice treated with DMA only and DMA + radiation, rather than the appearance of finger-like projections, vesicles were observed. This indicated a decrease in radiation-induced migration upon DMA treatment.

Radiation therapy in head and neck cancer patients causes salivary gland dysfunction. Patients with xerostomia experience symptoms such as oral discomfort or pain, resulting in difficulties swallowing and an increased risk of dental caries or oral infection [51]. Here, we demonstrated that the effect of DMA on maintaining the integrity of the salivary gland was correlated with the expression of AQP5 on the apical surface of the gland. The saliva produced in DMA-treated mice with or without radiation showed a two-fold increase, whereas amifostine showed a one-fold increase in the HK-1 xenograft mouse model. In the PDX mouse model, the saliva produced in DMA-treated mice with or without radiation showed a more than four-fold increase compared to the radiation treatment group.

The current study proves that the radiation response of nasopharyngeal and tongue squamous cell carcinoma tumor-carrying mice was significantly enhanced in the presence of DMA at a radiation dose above 4 Gy per fraction threshold. Our in vivo experiments using the HK-1 xenograft and PDX mouse model suggest that combining DMA with radiation improves animal survival compared to radiation alone.

The histopathological evaluation of the tissue samples pretreated with DMA before radiation showed a reduced level of tissue injury with no adverse effects in the spleen and small intestine of the mice. Pathological changes were observed in lungs marked with the degeneration of the bronchiolar epithelium and focal necrosis in the radiation group only. The immunoblot analysis of tumor tissue was in concordance with the immunohistochemical findings of AQP5 expression. DMA treatment before radiation resulted in a significant increase in the expression of the AQP5 protein. The above findings indicate that DMA treatment preserved the integrity of the glandular lobule following irradiation.

Ki-67 is a nuclear protein, and its expression is widely used as a prognostic marker in cancer [52]. Tissue sections from the animals pretreated with DMA at a dose of 50 mg/kg following radiation exposure at 4 Gy also showed an elevation in the number of Ki-67-stained cells in the spleen, intestine, and lungs. Based on these results, we postulate that DMA treatment in head and neck cancer patients before radiation can prevent radiation-induced tissue injury and promote cell proliferation in normal tissues.

Head and neck cancers are highly vascular tumors with a tendency to metastasize. They express a wide range of integrin receptors. αvβ3 and αvβ5 integrin expression is significantly higher in oral tumor cells [53] and tongue squamous cell carcinoma than in epithelium cells in normal tissues. Its expression is associated with lymphatic metastasis and angiogenesis [54]. DMA treatment with or without radiation significantly decreased the expression of integrin αvβ3, which could point towards DMA as negatively affecting tumor growth and migration in the patient-derived xenograft model. This was corroborated by the whole-body SPECT imaging of the patient-derived xenograft model of tongue squamous carcinoma using 100μCi^99m^Tc-labeled RGD peptides.

CD44 is widely used as a cancer stem cell marker [55]. It is a strong indicator of cell migration and tumor metastasis. The expression of CD44 was lower in the DMA and DMA + 4 Gy groups of animals than the high level of expression observed in the radiation treatment group.

Earlier, it was reported that the mitogen signaling pathway (Ras/Raf/MEK/ERK) can mediate not only cell proliferation and survival [56] but also cell cycle arrest and death in different cell types [57]. The Raf/MEK/ERK pathway is known for its ability to promote cellular proliferation and survival. In a contrast to this, a significant amount of evidence suggests that the Ras/Raf/MEK/ERK pathway can also be mediated by the apoptosis pathway [58]. It had also been well demonstrated that the overexpression of ERK1/2 can switch C-Raf-induced growth arrest responses to caspase-dependent apoptotic death responses [59]. In the present study, however, we provide evidence that the activation of ERK is important for the induction of apoptosis in PDX tumor tissue upon DMA treatment. DMA treatment resulted in the increased expression of the Ras protein, which led to the activation of ERK and caspase leading to programmed cell death. Treatment with AMG510 resulted in the suppression of ERK phosphorylation, leading to apoptosis. It was observed that the inhibition of KRAS and ERK1/2 together in tumor cells synergistically increased the cytotoxicity causing cell death. In the annexin assay, the percentage of early and late apoptotic cells was substantially increased in the DMA and DMA + 4 Gy-treated HK-1 cells compared toAMG510 and AMG510 + 4 Gy-treated cells [27,28].

The overexpression of cyclin B1 is associated with transformed cells and is a marker of poor prognosis for a variety of cancers [60]. Our results in head and neck PDX tumor tissue indicate a positive correlation between the levels of cyclin B1 protein and apoptosis induced upon DMA treatment.

The biochemical profile of the blood of DMA-treated mice showed a reduction in the levels of AST and ALP enzymes. As per the literature, the AST levels of cancer patients have been found to be elevated, resulting in severe necrosis and neoplasia. Our results suggest that AST levels were substantially reduced in the DMA and DMA + radiation-treated mice compared to untreated controls, which suggests that DMA prevents hepatotoxicity and reduces cellular damage. High serum ALP is used as a diagnostic marker and is associated with a variety of diseases. The DMA + radiation treatment reduced the level of ALP compared to that of the control mice.

## 5. Conclusions

Conclusively, our findings emphasize that the administration of DMA in OSCC patient-derived xenografts not only prevents radiation-induced damage but also considerably slows tumor progression (Figure 9). Although radiation and surgery are well-established therapies for HNSCC, the recurrence of disease, death of 50% of patients, and xerostomia ultimately ensues. The molecular mechanism of DMA suggests that reduced migration and metastasis and increased overall survival in patients will eventually lead to more effective treatment strategies to improve outcomes for patients. We expect to use our validated PDX mouse model of OSCC for screening new anticancer agents for the therapy of this devastating disease.

## Figures and Tables

**Figure 1 cancers-14-04908-f001:**
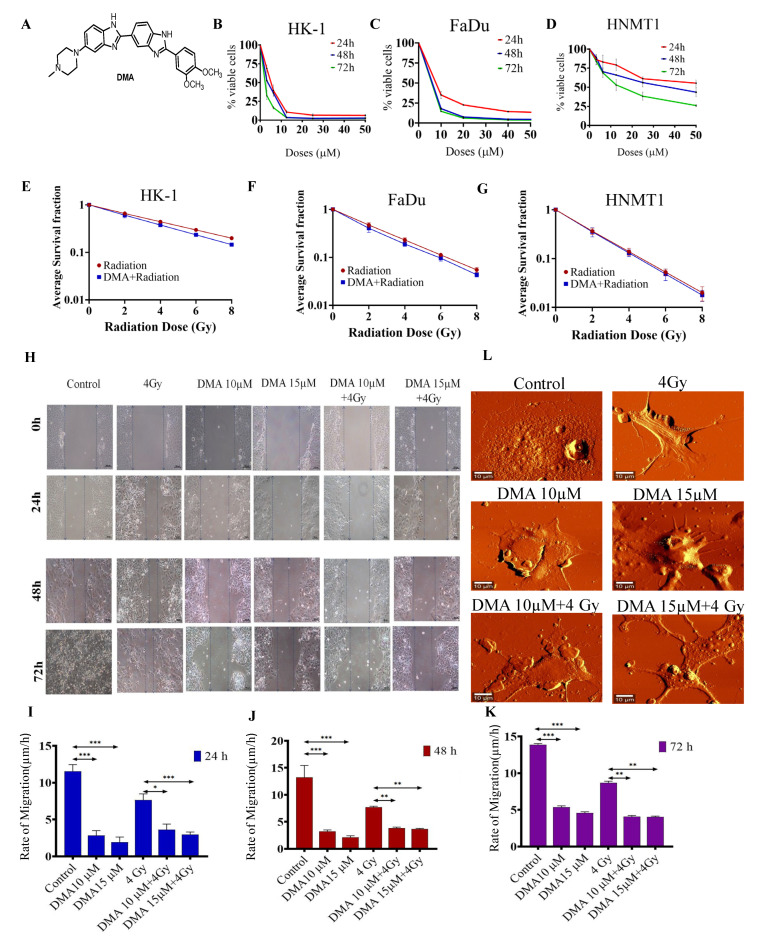
DMA treatment induced the killing of cancer cells and decreased cell migration in head and neck cell lines and HNMT1 primary cells. (**A**) Chemical structure of DMA. Cytotoxicity of DMA on (**B**) HK-1, (**C**) FaDu, and (**D**) tongue squamous cell carcinoma patient-derived primary (HNMT1) cells determined through MTT assay. (**E**–**G**) Clonogenic survival curves of HK-1, FaDu, and HNMT1 cells treated with DMA at 6, 11, and 34 µm for 2 h before X-ray irradiation at 2, 4, 6, and 8 Gy, respectively. (**H**) Time course of scratch closures; HK-1 monolayers were mechanically scratched with a sterile pipette tip following treatment with 10 or 15 µM DMA, 4 Gy radiation, and DMA + 4 Gy radiation. (**I**–**K**) The relative size of the open scratch area was measured at 24, 48, and 72 h following treatment with 10 or 15 µM DMA, 4 Gy radiation, and DMA + 4 Gy radiation. (**L**) Representative atomic force microscopy images of morphological changes in untreated HK-1 cells, cells treated with 10 or 15 µM DMA for 2 h, 4 Gy radiation, and DMA treatment for 2 h followed by 4 Gy radiation (DMA + 4 Gy). Scale bar = 10 µm. Data are shown as means ± SEM of three replicates. * *p* value < 0.05, ** *p* < 0.005 and *** *p* value < 0.0005.

**Figure 2 cancers-14-04908-f002:**
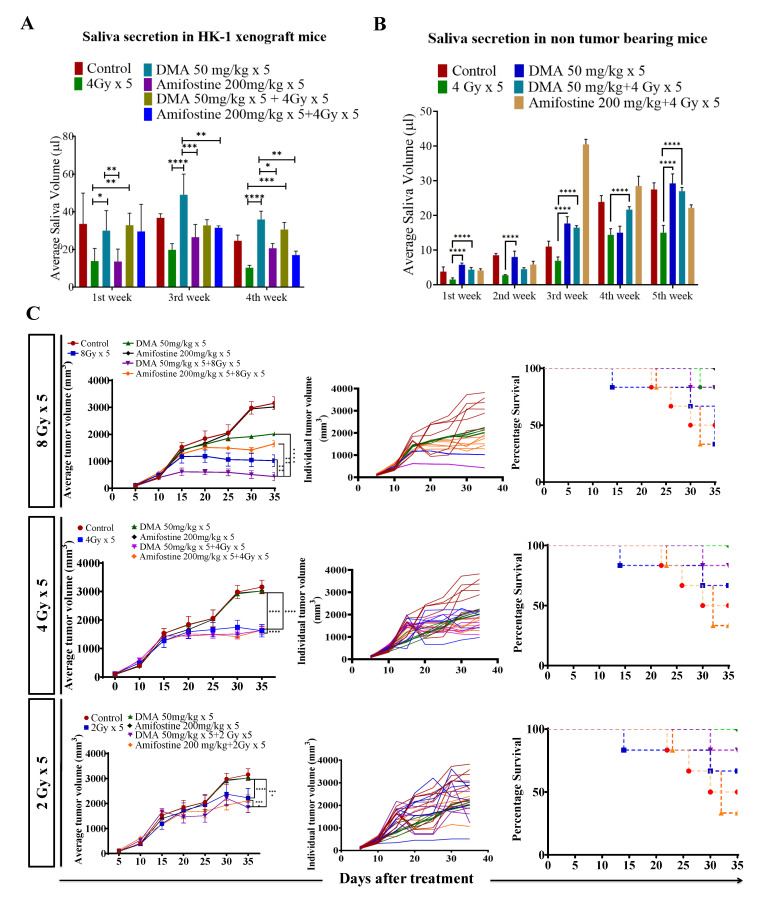
DMA restores saliva secretion and causes the enhancement of radiation-induced tumor growth delay (TGD) with an increasing radiation dose per fraction in the HK-1 xenograft mouse model. The animals were divided into four groups: control (tumor-bearing), radiation-treated, DMA-treated, and DMA + radiation-treated mice. Measurement of stimulated saliva secretion in (**A**) HK-1 xenograft-bearing nude mice for the first, third, and fourth week and (**B**) non-tumor-bearing NCr-*FoxN1^nu^* nude mice. (**C**) HK-1 xenograft was treated with IR schedules of 8 Gy × 5, 4 Gy × 5, and 2 Gy × 5 fractions alone and in combination with DMA given once per day for 5 days, 2 h before irradiation. Average tumor volumes (left column), individual tumor volumes (middle column), and Kaplan–Meier analysis for survival are shown. All animal cohorts contained n = 6 animals per group. Data were analyzed using two-way ANOVA followed by Tukey’s multiple comparisons. *p*-values at * < 0.05, ** < 0.005, *** < 0.0005, and **** < 0.00005 indicate statistical significance.

**Figure 3 cancers-14-04908-f003:**
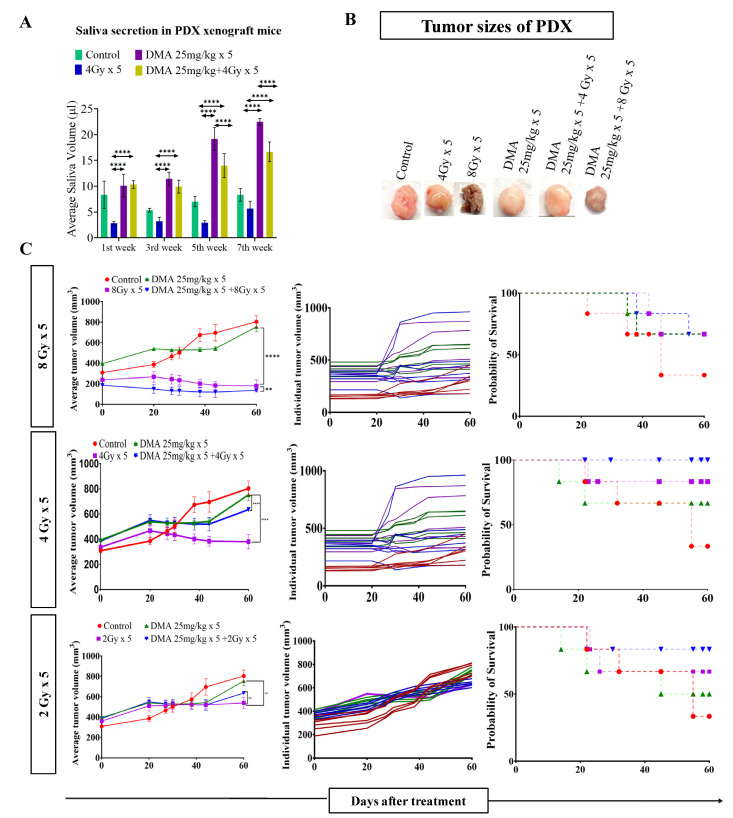
DMA restores saliva secretion and causes the enhancement of radiation-induced tumor growth delay (TGD) with an increasing radiation dose per fraction in the PDX mouse model. Measurement of stimulated saliva secretion in (**A**) the first, third, fifth, and seventh week in the PDX model with (**B**) a pictorial depiction of the tumor sizes after the completion of the study (after 60 days from the start of the treatment). (**C**) The PDX was treated with IR schedules of 8 Gy × 5, 4 Gy × 5, or 2 Gy × 5 fractions alone and in combination with DMA given once per day for 5 days, 2 h before irradiation. Average tumor volumes (left column), individual tumor volumes (middle column), and Kaplan–Meier analysis for survival are shown. All animal cohorts contained n = 6 animals per group. Data were analyzed using two-way ANOVA followed by Tukey’s multiple comparisons. *p*-values at ** < 0.005 and **** < 0.00005 indicate statistical significance.

**Figure 4 cancers-14-04908-f004:**
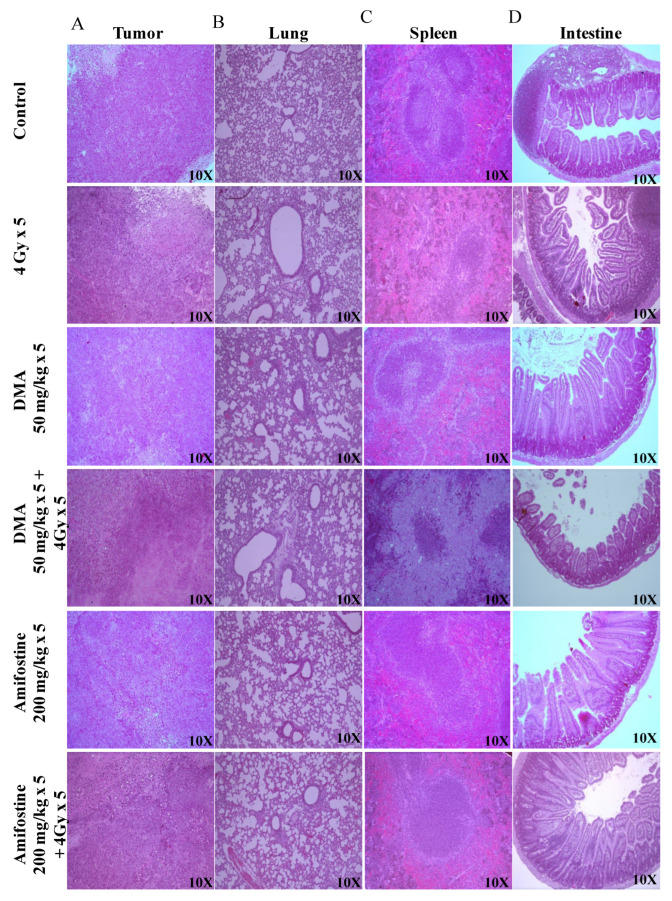
DMA ameliorates radiation-induced damage in normal tissues compared to tumor tissue in HK-1 xenograft. Histological analysis of radiation injury via H&E staining of (**A**) tumor, (**B**) lungs, (**C**) spleen, and (**D**) intestine with and without DMA treatment (50 mg/kg *i.v.*) at a magnification of 100×. Images are representative of three tissue samples and three areas per slide. The H&E scoring of the histological analysis of normal tissue is tabulated in Appendix A.

**Figure 5 cancers-14-04908-f005:**
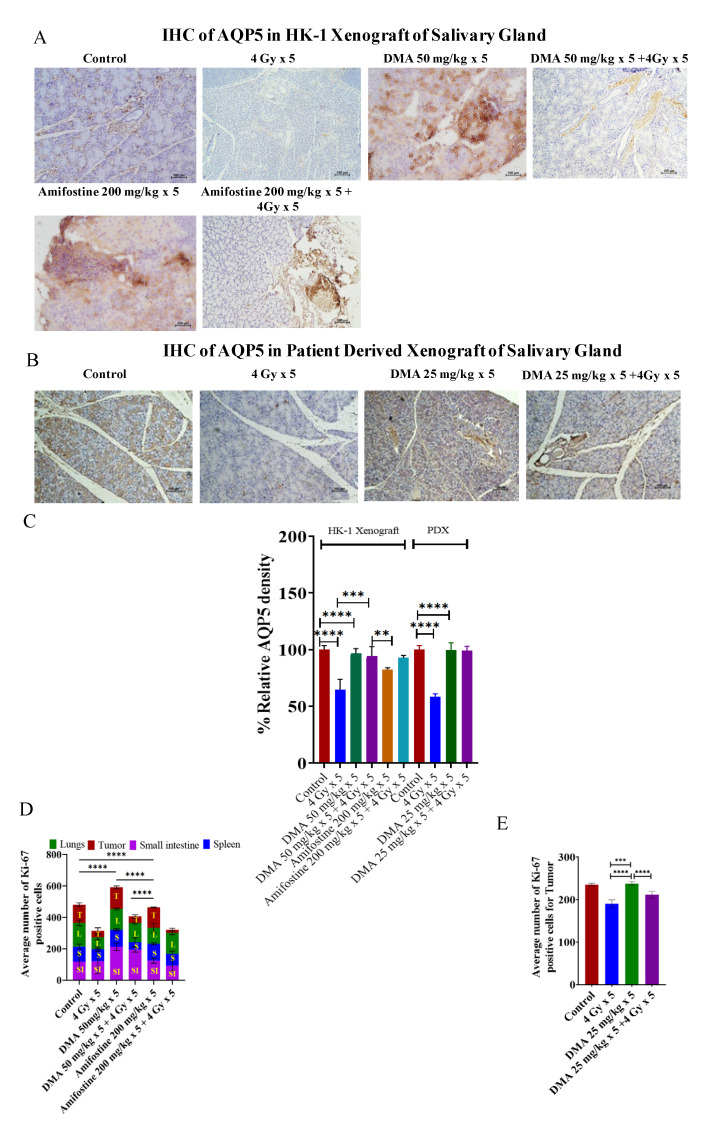
Expression of AQP5 in the salivary gland of xenograft models. Representative images depicting immunostaining for AQP5 in (**A**) HK-1 xenograft and (**B**) PDX salivary gland sections; images are representative of three tissue samples. Scale bar = 100 µm. Bar graphs showing the immunostaining quantification of (**C**) AQP5 for HK-1 and PDX mouse models; the average number of Ki-67-positive cells for the quantification of (**D**) HK-1 and (**E**) PDX models was performed using15 random histological slides. The brown color represents the positive staining of AQP5 with blue counterstaining. *p*-values at ** < 0.005. *** < 0.0005 and **** < 0.00005 indicate statistical significance.

**Figure 6 cancers-14-04908-f006:**
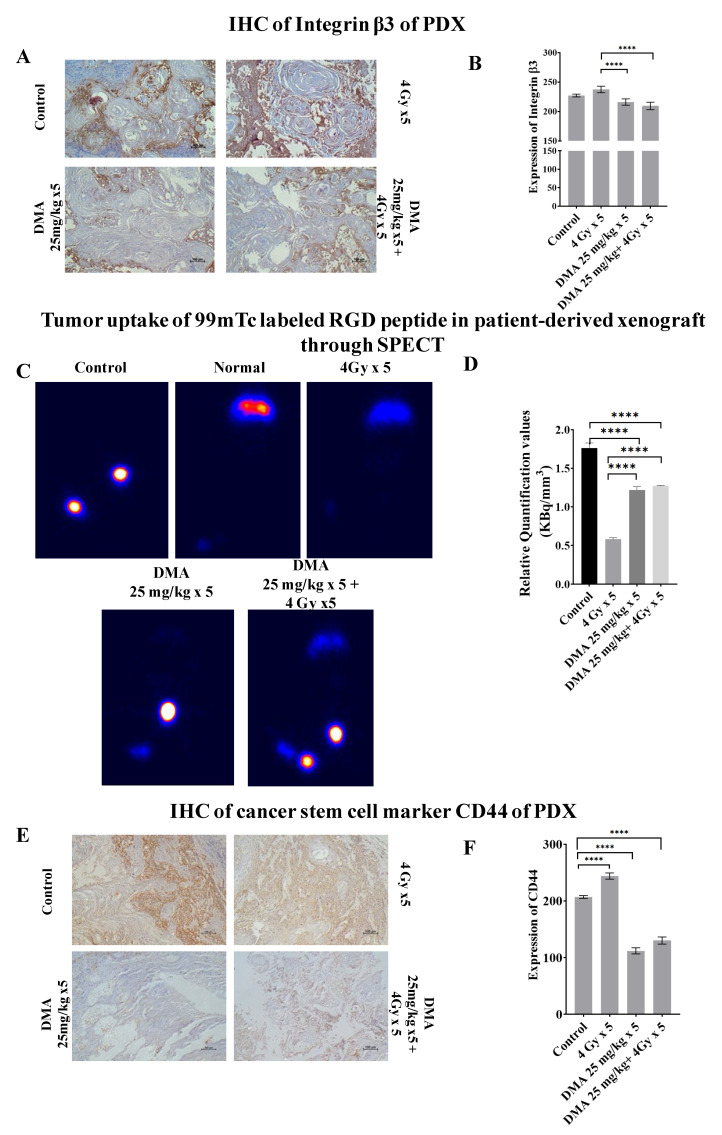
Abrogation of integrin and CD44 expression results in reduced tumor metastases. (**A**) Representative images depicting immunostaining for integrin β3. (**B**) Bar graph showing immunostaining quantification of integrin β3. (**C**) Comparison of tumor uptake of ^99m^Tc-labeled RGD peptide in the patient-derived xenograft using SPECT for control (no treatment given to mice with tumor), normal (no treatment given to mice without tumor), radiation alone (4 Gy × 5), DMA (25 mg/kg × 5), and DMA treatment for 2 h followed by radiation (DMA 25 mg/kg + 4 Gy × 5). (**D**) Bar graph showing relative quantification valueof tumor uptake for ^99m^Tc-labeled RGD normalized to respective tumor volume. (**E**) Representative images depicting immunostaining for CD44 on PDX mouse model tumor sections. The images are representative of three tissue samples. Scale bar = 100 µm. (**F**) Bar graph for the quantification of CD44 expression. *p*-values at **** < 0.00005 indicate statistical significance.

**Figure 7 cancers-14-04908-f007:**
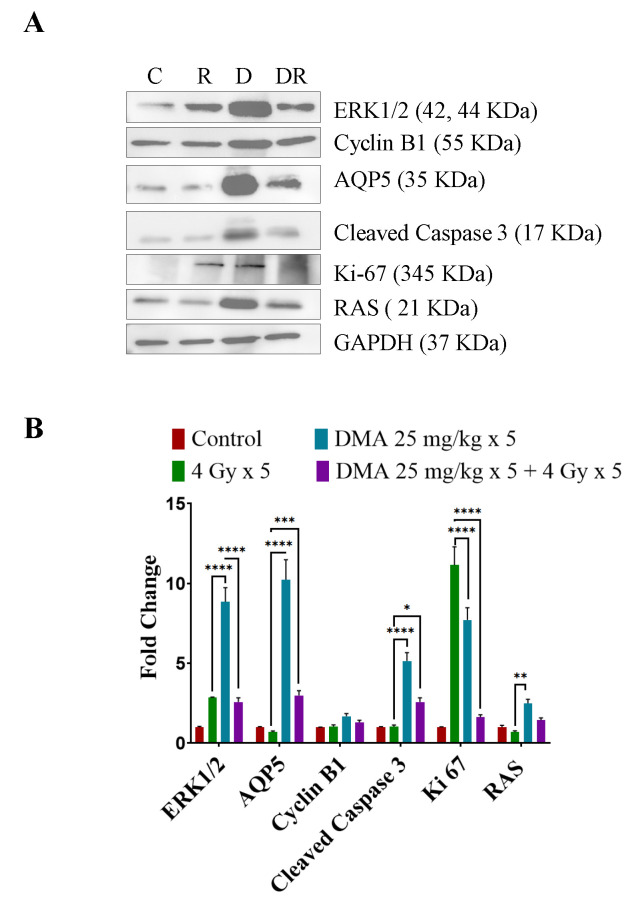
Differential regulation of downstream effector proteins in response to DMA treatment. (**A**) Immunoblot analysis of proteins involved in the Ras/Raf/MEK/ERK pathway in tumors derived from the tongue squamous cell carcinoma patient-derived xenograft at the end of the study (60 days) from the control (C), 4 Gy × 5 (R), DMA 25 mg/kg × 5 (D), and DMA 25 mg/kg × 5 + 4 Gy × 5 (DR) groups. (**B**) Densitometric analysis of representative Western blot. Band intensities were normalized to those of the normal form of each protein, GAPDH. *p*-values at * < 0.05, ** < 0.005, *** < 0.0005 and **** < 0.00005 indicate statistical significance.

**Figure 8 cancers-14-04908-f008:**
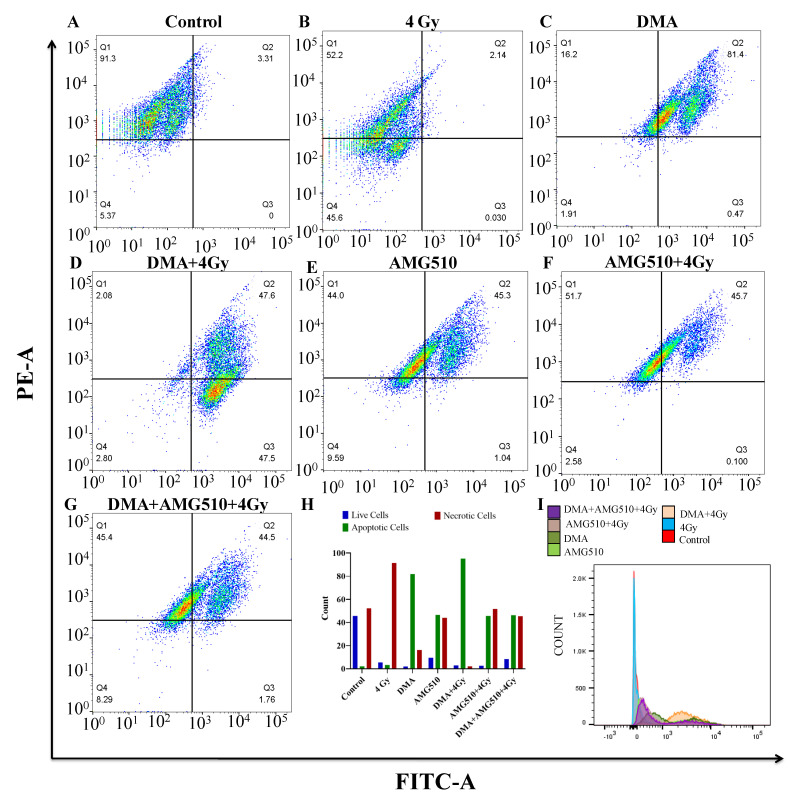
DMA induces apoptosis in nasopharyngeal HK-1 cells. Contour diagram of the annexin V-FITC/PI flow cytometry of nasopharyngeal HK-1 cells: (**A**) control, (**B**) radiation (4 Gy), (**C**) DMA (6.25 µM), (**D**) DMA + radiation (6.25 µM + 4 Gy), (**E**) AMG 510 (10 µM), (**F**) AMG 510 + radiation (10 µM + 4 Gy), and (**G**) DMA + AMG 510 + radiation (6.25 µM + 10 µM + 4 Gy). The lower left quadrant (Q4) of the cytograms shows the viable cells, which excluded PI and were negative for annexin V-FITC binding. The upper right quadrant represents late apoptotic cells, positive for annexin V-FITC binding and showing PI uptake (Q2). The lower right quadrant (Q3) represents the apoptotic cells, annexin V-FITC positive and PI negative, demonstrating annexin V binding and cytoplasmic membrane integrity. (**H**) Quantitative percentages (count %) of live, apoptotic (early and late apoptotic cells), and necrotic cells. (**I**) Histogram of the population of cells (count %) undergoing apoptosis stained with annexin V.

**Figure 9 cancers-14-04908-f009:**
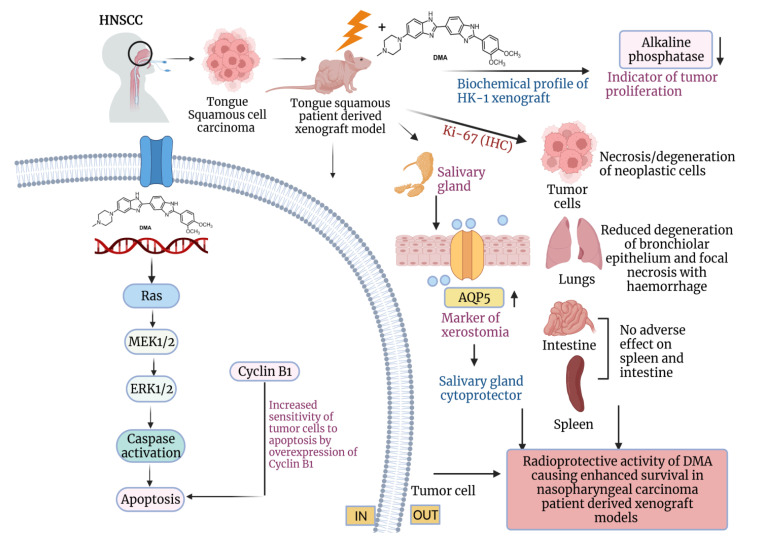
Schematic representation of the radioprotection of normal cells and tumor ablation by DMA in xenograft mouse model. The proposed model shows that DMA with irradiation activates the RAS/MEK pathway, leading to increased apoptosis in PDX tumor tissues. Further, the reduced levels of alkaline phosphatase, restoration of aquaporin water channels on salivary glands, and reduced Ki-67 levels in tumor cells, as well as normal tissues, contribute to the radioprotective potential of DMA in normal cells, which leads to enhanced survival in PDX models.

## Data Availability

The data presented in this study is available in this article (and Appendix A).

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
