# Peer review of "DMA, a Small Molecule, Increases Median Survival and Reduces Radiation-Induced Xerostomia via the Activation of the ERK1/2 Pathway in Oral Squamous Cell Carcinoma"

_cancers, 2022, doi:10.3390/cancers14194908_

Round 1
Reviewer 1 Report
The authors revised their manuscript to answer the reviewers' comments.
Although the quality of their presentation has been improved, errors are still observed through the paper. They have to be revised again completely and absolutely.
Line 96, cell line-derived
Line 328, "tongue squamous patient" is still strange.
Figure 3B, Was this photo taken at the 60th day? It should be described.
Line 423, "stainingof" should be "staining of".
Line 424, "protocoldescribed" should be "protocol described".
Line 425, model"section.. Excess of period.
Line 484-485, 211.6±8.2 is repeated.
Line 495. "tongue squamous patient-derived xenograft" should be "patient-derived xenograft".
Line 504, The value of SPECT for normal should be described.
Figure 6B (99mTc SPECT) must demonstrate the function of salivary glands. The meaning of 99mTc SPECT have to be explained for readers who don't know about 99mTc SPECT.
Figure 7B, There is no bar to show standard deviation. If they performed densitometric analysis only one time, statistical analyses is impossible.
Figure 7A and B, The bot of ERK1/2 does not seem to be increased comparing R and DR.
Line 550-551, Early apoptotic and late apoptotic cell population is really 3.31% for radiation (4Gy)? It is totally different from Figure 8H. The percentages in line 550-551 seem to represent only early apoptotic cell population.
Line 657, cell death. . Excess of period.
Round 2
Reviewer 1 Report
The authors addressed all of the comments I asked. The quality of their report has been well refined for readers' understanding. Now, their research and findings are very interesting for head and neck oncologists. I respect the authors' effort to complete this report.